# Does the Polypill Improve Patient Adherence Compared to Its Individual Formulations? A Systematic Review

**DOI:** 10.3390/pharmaceutics12020190

**Published:** 2020-02-22

**Authors:** Ana Baumgartner, Katarina Drame, Stijn Geutjens, Marja Airaksinen

**Affiliations:** 1Division of Pharmacology and Pharmacotherapy, Faculty of Pharmacy, University of Helsinki, 00014 Helsinki, Finland; katarina.drame@gmail.com (K.D.); stijngeutjens@gmail.com (S.G.); marjaairaksinen@gmail.com (M.A.); 2Faculty of Pharmacy, University of Ljubljana, 1000 Ljubljana, Slovenia; 3Faculty of Pharmaceutical Sciences, Katholieke Universiteit Leuven, 3000 Leuven, Belgium

**Keywords:** polypill, fixed-dose combination, adherence, systematic review

## Abstract

Many patients, especially those with a high pill burden and multiple chronic illnesses, are less adherent to medication. In medication treatments utilizing polypills, this problem might be diminished since multiple drugs are fused into one formulation and, therefore, the therapy regimen is simplified. This systematic review summarized evidence to assess the effect of polypills on medication adherence. The following databases were searched for articles published between 1 January 2000, and 14 May 2019: PubMed, Web of Science, Cochrane Library, and Scopus. Medication adherence was the only outcome assessed, regardless of the method of measuring it. Sixty-seven original peer-reviewed articles were selected. Adherence to polypill regimens was significantly higher in 56 articles (84%) compared to multiple pill regimens. This finding was also supported by the results of 13 out of 17 selected previously published systematic reviews and meta-analyses dealing with this topic. Adherence can be improved through the formulation of polypills, which is probably why the interest in researching them is growing. There are many polypills on the market, but the adherence studies so far focused mainly on a small range of medical conditions.

## 1. Introduction

Poor medication adherence is a widespread and unresolved challenge among patients [1]. Only half of the prescribed doses are taken, and many patients stop their treatment before the planned end of the therapy [1,2,3]. Several factors contribute to low adherence rates, such as ineffective communication between the patient and the physician, or patients perceive their treatment as unnecessary. Patients may also think the benefits of their pharmacotherapy do not outweigh its adverse effects, or they simply forget [4]. This very often results in complications, extra healthcare costs, side effects, and therapeutic failures. Therefore, improving adherence is a crucial factor in increasing the likelihood of positive therapeutic outcomes.

Patients with chronic illnesses must quite often take multiple pills every day for months or even years, which will eventually result in less adherence to their medication [5]. This occurs especially in cardiovascular diseases (CVD), where patients do not feel the symptoms of their disease in the short term, and it is easier for them to forget to take their medicines [6]. The same goes for diabetes patients; in mild forms, diabetes does not cause serious complications, and patients do not feel ill; thus, they tend to forget their medication [7].

Polypills are a technological innovation that is expected to improve adherence by simplifying the pharmacotherapy regimen [2,5]. The concept of the polypill, very often referred to as a fixed-dose combination (FDC), is quite simple. Instead of taking two or more pills (each containing one active ingredient), multiple drugs are combined into one formulation [6,8,9]. It is generally thought that taking fewer pills will lead to better adherence [2]. This systematic review examines the evidence for that idea and assesses the evidence of the effects of a reduced pill burden on medication adherence.

## 2. Materials and Methods

### 2.1. Search Strategy

This systematic review is focused on articles concerning fixed-dose combinations (FDCs), also known as polypills, in comparison to their separate drug formulations (multiple tablets, free-pill combinations). It does not matter how many drugs are combined in a certain formulation.

The method followed the Preferred Reporting Items for Systematic Reviews and Meta-Analyses (PRISMA) statement [10]. The search was done in May 2019, and it covered the following databases: PubMed, Web of Science, Cochrane Library, and Scopus. After screening all titles and articles, the reference lists of selected articles were used to identify additional relevant studies.

In all four databases, the following selection of keywords was applied: (compliance OR adherence OR non-compliance OR non-adherence OR noncompliance OR nonadherence) AND ((fixed NEAR/1 combination*) OR single-pill* OR single-tablet* OR polypill* OR “combination pill*”). The principle behind this selection was to make an extensive search that would cover only the relevant articles by using as many synonyms and antonyms for two terms related to the aim of our study: polypills and adherence. The search included a language filter, which showed only articles written in English. Furthermore, a time-span filter was used, which included only articles published since 1 January 2000.

### 2.2. Inclusion and Exclusion Criteria

Articles were included in the systematic review if they were either original peer-reviewed studies or systematic reviews and meta-analyses. Narrative reviews and conference abstracts were excluded. There were no restrictions concerning the type of patients, diseases, comorbidities, or drugs. Adherence was the only essential outcome measure for an article to be considered, regardless of how it was measured. Other outcomes were not assessed. All articles needed to have a comparison between low and high pill burden groups, meaning that one group had to take more pills than another group. This was possible either with control groups (longitudinal, controlled) or when observing one group with patients who changed their pill burden over time (longitudinal, uncontrolled). It was also necessary that the articles dealt with solid dosage formulations rather than with any other dosage form.

Additionally, articles dealing with persistence instead of adherence were excluded. The definitions of these two terms are vague since they stand for a similar phenomenon and, thus, they tend to overlap in different literature. However, for this study, only articles measuring adherence were included, and the ones that clearly stated that they dealt only with persistence were excluded.

### 2.3. Study Selection and Data Extraction

Both researchers (A.B. and K.D.) searched for the articles separately to make the most credible and objective article selection. Their findings were then compared, and discussions were held until a final decision about included articles was reached.

Key information about all relevant studies was extracted from the articles. For original peer-reviewed studies, the extracted information included author of the study, year of publication, study country, design, setting, aim and population, disease in question, follow-up period, adherence measures, main outcomes, number and international nonproprietary names (INN) of active ingredients used in the study, and their dosage (if given). For systematic reviews and meta-analyses, the extracted data covered the author of the study, year of publication, medical condition in focus, study aim, number of included original studies, and main results. For further working progress, articles were grouped by (i) article type, that is, original studies separately from systematic reviews and meta-analyses, and (ii) the type of disease they were dealing with.

### 2.4. Quality Assessment

Quality assessment of the included studies was systematically done using Cochrane Collaboration’s risk of bias assessment tool for randomized controlled trials [11] and the Newcastle–Ottawa scale for cohort studies [12]. As adherence was the only outcome of interest, assessment of how any other outcomes were dealt with was ignored.

Cochrane Collaboration’s risk of bias tool measures risk of bias in randomized controlled trials through seven domains: sequence generation and allocation concealment (selection bias), blinding of participants (performance bias), blinding of outcome assessment (detection bias), incomplete outcome data (attrition bias), selective reporting (reporting bias), and other bias. Each domain is assessed as having high, low, or unclear risk of bias. Studies with low risk of bias for all criteria were considered to be of low risk, studies with low or unclear risk of bias were considered medium risk, and studies with at least one domain assessed as high risk of bias were considered as having an overall high risk of bias [11]. However, as all randomized controlled trials were inherently open-label, meaning that blinding of participants was impossible due to the nature of the intervention, this domain was always assessed as having a high risk of bias and, therefore, omitted from the overall evaluation of studies.

The Newcastle-Ottawa scale was used for observational studies; it measures quality through three main domains: selection, comparability, and outcome. Each study was awarded a maximum of nine stars, depending on whether it reached certain standards within these domains (maximum four stars for selection, two stars for comparability, and three stars for outcome) [12]. Studies with 0–4 stars were considered as low quality, studies with 5–7 stars were considered as medium quality, and studies having eight or nine stars were considered as high quality.

## 3. Results

The database search yielded 5170 records, of which 2287 were screened after removal of duplicates and inclusion of time-span and language filters. After the inclusion and exclusion criteria were applied to screen the records, 84 articles were included in our systematic review; 67 of them were original peer-reviewed studies and 17 were systematic reviews and/or meta-analyses. For the flow chart of the article selection process, see Figure 1.

### 3.1. Included Systematic Reviews and Meta-Analyses (n = 17)

After the selection process of eligible studies, 17 systematic reviews and/or meta-analyses out of 136 articles were included in this systematic review (Figure 1, Table A1, Appendix A). Of these studies, eight were meta-analyses (47%) [13,14,15,16,17,18,19,20], two were systematic reviews (12%) [21,22], and seven were defined as a systematic review with meta-analysis (41%) [23,24,25,26,27,28,29]. The most common systematically reviewed medical condition was hypertension (*n* = 5, 29%) [14,15,16,25,26], followed by studies dealing with CVDs in general (*n* = 3, 18%) [17,18,21], human immunodeficiency virus (HIV) (*n* = 4, 24%) [19,20,24,27], diabetes (*n* = 2, 12%) [22,28], and tuberculosis (*n* = 1, 6%) [23] (Table 1). Two studies examined the effect of polypills in several medical conditions [13,29]. Thirteen of the selected articles (76%) favored therapy with FDC over separate-pill therapy regimens [13,14,15,16,18,19,20,21,22,24,25,27,28].

However, some overlap of the articles that these studies investigated was found. For example, Selak et al. [17] and Webster et al. [18] included the same studies, which were also included by Bahiru et al. [21] Furthermore, there is much overlap between studies conducted by Gupta et al. [16], Kawalec et al. [25], Sherrill et al. [15], and Du et al. [14]. Both studies conducted by Clay et al. [20,24] share some of the included articles as well. For a visual representation of the overlap of the studies included in the abovementioned systematic reviews and meta-analyses, please see Figure 2.

Some of the studies investigated in one or more of the previously published systematic reviews and meta-analyses were also included in our systematic review since they fit the inclusion criteria [5,30,31,32,33,34,35,36,37,38,39,40,41,42,43,44,45,46,47,48,49,50,51,52,53,54,55,56,57].

### 3.2. Included Original Peer-Reviewed Studies (n = 67)

Altogether, 67 original peer-reviewed studies out of a total of 5170 articles met the inclusion criteria and were included in our study (Figure 1). Of the 67 articles, 31 (46%) were related to hypertension (HT), 14 (21%) were related to human immunodeficiency virus (HIV), 11 (16%) were related to cardiovascular disease (CVD), 10 (15%) were related to diabetes mellitus type II (DMII), and one dealt with lower urinary tract symptoms associated with benign prostatic hyperplasia (LUTS/BHP). Studies were conducted in different countries worldwide; some of them even included more than one country. Most of them (*n* = 36) were conducted in the United States, and only one was carried out in South America, in two different countries. More details can be found in Table 2.

A summary of these 67 studies can be found in Table A2 (Appendix B), displaying author, year of publication, country of the study, study design, study aim, study setting and follow-up period, study population, outcome measures, and results. 

Most of the studies (*n* = 41; 61%) examined the effects of treatment with polypills, where only two drugs were combined (see Table 2 and Table 3). However, in HIV therapy, the use of three-drug formulations was dominant over any other (seven out of 14 studies). There were no data on combining more than five drugs into one formulation. Additional information about the combinations of active ingredients in polypills can be found in Table A3 (Appendix C).

### 3.3. Adherence Measures Used in the Studies (n = 67)

Table 4 summarizes methods for measuring adherence that were used in the selected articles (*n* = 67). Most of the studies (*n* = 62; 93%) relied only on one method; however, five studies [35,50,59,79,82] combined two different methods to assess medication adherence. The applied methods could be divided into two broad categories: subjective (e.g., patient interviews and self-reporting) and indirect (e.g., pill counts, methods using prescription fills, electronic monitoring) [75]. Some of the methods are more general and applicable to more cases, whereas some were used only in a specific study. The most commonly used measure was medical possession ratio (MPR; *n* = 30, 45%), followed by proportion of days covered (PDC; *n* = 21, 31%).

### 3.4. Adherence Outcome

All studies had one or more groups that received more pills than their control groups (Table A2, Appendix B). Those groups could be either a cohort of the same group or a comparison between two different groups. In most cases, the control group was a group of patients on the usual therapy (multiple pills). The test subjects received exactly the same active ingredients as the control group, but in a single formulation [31,32,33,35,41,42,43,54,59,61,67,69,70,73,81,85,89]; alternatively, the test group and the control group were not necessarily receiving the same drugs, but they simply had a different pill burden [5,30,34,36,37,38,39,40,44,45,46,47,48,49,50,51,52,53,55,56,57,58,60,62,64,65,66,68,71,72,74,75,76,77,78,79,80,82,83,84,86,87,88,90,91,92,93,94,95].

The main interest of this review is how the pill burden is associated with patient adherence. In 56 out of 67 studies (84%), there was a significant difference in adherence between the test and control group (Table 5). In seven studies (10%), the difference between both groups was insignificant. In only two studies (3%), both opposite outcomes (improved and decreased adherence in the test group, depending on the treatment situation before the study index date) were reported [61,80]. See Table 5 for a summary of the outcomes.

### 3.5. Quality Assessment of the Included Studies

Figure 3 shows the risk of bias summary for all seven randomized controlled trials [35,45,48,52,59,75,88]. As previously noted, blinding of participants was impossible due to the nature of the intervention, that is, different pill burden, resulting in high risk of performance bias in all studies. Thus, it was decided to be omitted from the overall risk of bias assessment. Based on the previously determined criteria, two RCTs reached standards for having an overall low risk of bias [48,52], two studies reached standards for having a medium risk of bias [35,59], and three studies were considered as having a high overall risk of bias [45,75,88].

Except for Matsumura et al. [75], which reported on the insignificant difference in adherence between polypills and multipill therapy and which was assessed as having a high risk of bias, all the other RCTs (*n* = 6, 86%) showed improved adherence when using polypills compared to multipill therapy.

From 60 of the included observational studies, 39 (65%) were assigned eight or nine stars according to the Newcastle–Ottawa rating and were, thus, considered as high-quality studies [5,33,36,37,41,42,43,44,46,47,50,51,53,54,55,56,58,61,62,63,64,65,67,71,72,74,76,78,79,80,81,82,83,84,86,89,90,91,94]. There were 19 studies (32%) that reached criteria for medium quality (six or seven stars) [30,31,32,34,38,39,40,49,57,60,68,69,70,77,85,87,92,93,95], and only two studies (3%) were considered poor quality, with both having five stars assigned [66,73].

In 50 out of a total of 60 observational studies (83%), adherence to the polypill was shown to be increased compared to multipill therapy. Of the high-quality studies, 31 out of 39 studies (79%) also showed this outcome, which does not differ importantly from findings from the total number of observational studies. Moreover, the ratio of studies with an insignificant difference in adherence to polypill and multipill therapy is very similar for high- and medium-quality studies (4/39 or 10% vs. 2/19 or 11%, respectively). For a visual representation showing the number of studies with a certain outcome concerning adherence per quality of cohort study, see Figure 4.

## 4. Discussion

The main strength of our study is the broad range of included original peer-reviewed studies and that no restrictions concerning the medical condition, type of patients, or adherence measures were used in the research. Based on this systematic review, there is a connection between pill burden and medication adherence in medical conditions such as hypertension, diabetes mellitus type 2, cardiovascular diseases, and HIV. This is reflected by the fact that, in 56 out of 67 examined studies (84%), patient adherence to single-pill fixed-dose combination therapy was significantly higher compared to free-dose combination therapy with multiple pills. Most of the 17 previously conducted meta-analyses and systematic reviews included in our research also suggested a positive effect of polypills on patient adherence. However, four out of 17 studies (24%) did not reach the same conclusion; either the findings were inconclusive [26,29] or FDCT was simply not shown to be superior to multipill therapy [17,23]. It has to be acknowledged, however, that the number of analyzed articles in these studies was either three [17,26], five [23], or six [29]; thus, they might not be highly representative.

Ten percent (*n* = 7) of the individual studies did not observe improved adherence in patients receiving polypill therapy [34,50,58,75,76,90,95]. The authors of these articles suggested the following methodological reasons for their results: (1) the number of participants was too small to obtain significant results [34]; (2) calculation of MPR was made alternatively and therapeutic or in-class switches were allowed for [58]; (3) the pill burden for some multipill therapy regimens was not high enough to have a significant influence on adherence [34]; (4) the study period was not long enough to detect differences between the polypill and multipill groups [75].

Interestingly, two out of 67 studies [61,80], dealing with CVD and HT, respectively, observed both positive and negative outcomes regarding the influence of FDCT on adherence. For the study, dealing with CVD [61], the article’s authors suggested that the reasons for decreased adherence in patients taking polypills were adverse events. These were supposed to be falsely attributed to an active ingredient, which the patients in question were not receiving before the start of the study [61]. Authors of the other study dealing with HT, however, suggested a different reason for decreased adherence [80]. According to them, patients who were highly adherent to their previous treatment with free-combined antihypertensive drugs may not have been taking both of their antihypertensive medications at the same time and as prescribed. Hence, although they were switched to the equivalent FDC, their blood pressure lowered too much; therefore, they reduced the dose of FDCs on their own [80].

To our knowledge, no other previous systematic review in polypills and adherence covered as many original peer-reviewed studies and such a broad range of medical conditions as this. Our findings indicate that the rate at which polypill therapy is associated with higher adherence varies among medical conditions. In most of the studies on CVD, HT, and DMII therapies, adherence increased in patients with polypill therapy; however, in studies on HIV or LUTS therapies, no difference was observed in four out of 15, that is, 27% of the studies. These differences in results can be partly explained by the methodological issues already discussed above. Further research on diseases other than CVD, HT, DMII, and HIV is needed to get a better understanding of whether and how the medical condition influences the impact of reduced pill burden on adherence.

The research mainly revolves around cardiovascular polypills; the reason for this is probably the abundance of patients suffering from CVD and HT [96]. Despite fewer studies on polypills for diabetes and HIV and one for lower urinary tract symptoms associated with benign prostatic hyperplasia, fixed-dose combination therapy was not introduced to other diseases in terms of its potential to influence adherence. As this literature review shows promising results for polypills with 56 out of 67 included studies improving adherence, the research could be extended to a wider range of medical conditions and a wider range of populations and health systems, as well as beyond high-income countries. The current research on polypills is dominated by the research conducted in the USA, reflecting their situation.

There are also some limitations concerning this systematic review. The first one is related to the methodological quality of the selected studies. The results would be more valid if more of the study designs were randomized controlled trials instead of retrospective and prospective cohort studies. Since the study design differed between articles, it was also not possible to assess study quality using only one universal method. Thus, two separate methods were used, one for randomized controlled trials [11] and the other for observational studies [12]. Consequently, it was not possible to make a joint summary of study quality assessment including all articles. Another limitation is related to the countries and the medications included in the studies. Specifically, every country has different public health concerns, as well as health systems, services, and finances, which influence medication practices. Since most of the reviewed studies were from high-income countries, particularly from the USA, a distorted image of the use of polypills in the rest of the world is possible. It must also be acknowledged that patient adherence is affected by many variables, such as patient age, medical condition, and clinical outcomes, which varied significantly in our selected articles. The assessment of these factors was not the aim of this review, but they could have significantly influenced our findings. Furthermore, due to the lack of articles regarding other diseases, it is not possible to conclude whether polypills are associated with an increase in medication adherence on a general level. This can become clear only when more studies regarding the effect of polypills on adherence in other diseases are conducted. Moreover, there is some overlapping among previously published systematic reviews and meta-analyses, as some articles were included in more than one of them, thereby giving those studies more emphasis.

Another issue that must be acknowledged as a possible limitation to our study is the diversity of methods for measuring adherence that were applied in the included studies. In 67 articles, 11 different adherence measures were used, which makes the results of the studies more difficult to compare, thereby adding a possible source of bias. The methods range, on the one hand, from assessing prescription claims and the number of the pills taken (indirect methods) to analyzing questionnaires and patient self-assessment, on the other hand (subjective methods) [75], all having specific advantages and limitations. For example, indirect methods are a very technical way of measuring adherence, but easy to apply to bigger datasets, which may explain their use in these studies. However, unlike self-assessment-based methods, they do not cover other aspects of a patient´s life that also influence adherence. Additionally, pill-count and prescription claim analyses do not guarantee that the patients were taking the prescribed medicines [40,49]. On the other hand, self-assessment is a very subjective way of measuring adherence and is often prone to over-reporting the actual state [45,87]. In none of the studies were direct methods of measuring adherence used (i.e., measuring blood concentration of the active ingredients), which are the most objective methods, but also the most difficult, time-consuming, and expensive, as well as being inconvenient for patients [48]. In the future, it might be useful to consider the aspect of how patients perceive taking one instead of multiple pills in the methods for assessing adherence. Finally, due to such substantial heterogeneity between studies in terms of study design and reporting on the outcome, a pooled estimate of the effect of the pill burden on adherence was not analyzed, which can be considered as one of the limitations as well.

Another issue that emerged while carrying out this systematic review relates to the vague use of the terms “adherence” and “persistence.” Certain methods (e.g., duration to treatment discontinuation) were defined as a measure of adherence in some of the studies, whereas the other studies stated it as a method of assessing therapy persistence. In the future, clearer definitions and distinction between these two terms and the methods used for measuring them should be made available to avoid misconceptions about the aims of the studies.

The years of publication of selected articles indicate that there was increased interest in polypills in the last years. Only seven of the selected articles (10%) were published before 2008, and 35 out of a total of 67 studies (52%) were published in the last six years. One of the reasons for this rising trend might be the fact that fixed-dose combination therapy shows promising results for improving patient adherence.

Most of today’s commercially available polypills are intended for the treatment of only one indication. However, since it was already established how beneficial FDCT can be for patients, another interesting concept that is not yet applied very often, but is worth considering, is combining drugs for different indications into one formulation. Between 2010 and 2015, two FDCs composed of active pharmaceutical ingredients (APIs) for different comorbid diseases were already approved, both without full clinical study data [97]. In the future, more emphasis could be given to such FDCs, since that would reduce pill burden even more and, hopefully, have an even greater effect on patient adherence.

Even though this systematic review shows one of the potential benefits of polypill therapy, some of its disadvantages should also be acknowledged. For example, if dosing titration is needed, fixed-dose combination therapy can be inflexible if the appropriate dosage is not available in the form of a polypill [98,99,100,101]. That could lead to exposure of patients to unnecessary therapy and even adverse effects without added benefits [100]. Furthermore, if adverse effects occur, it cannot be possible for the patient to determine which of the components is causing them [101]. Another possible issue is that polypill therapy may be more expensive than multipill therapy [66,67,98].

All in all, the evidence shown in this systematic review constitutes a base for possible advantages of polypill therapy over multipill therapy, at least in the investigated medical conditions, when tackling the widespread and alarming problem of patient adherence to medication. Thus, the role of polypills in clinical practice should not be neglected, even though their contribution to increasing adherence is only partial. There are many other patient- and system-related factors, such as patient age and socioeconomic status, health literacy, disease and medication beliefs, adverse effects, medical condition and its seriousness, treatment costs, and clinical outcomes, which also play a major role in achieving positive outcomes [48,52,61,80,84]. However, reducing the complexity of pill regimens, especially in diseases where the number of pills can seem overwhelming for patients, could at least partly lead to increased medication adherence and, therefore, also improved clinical outcomes. Nonetheless, to better understand the role of polypills in clinical practice, a higher number of long-term randomized controlled trials dealing with different medical conditions will be needed.

## 5. Conclusions

This systematic review shows a connection between pill burden and medication adherence. In most of the included studies, adherence to polypill therapy was significantly higher compared to multipill therapy. Our findings indicate that the rate at which fixed-dose combination therapy is associated with higher adherence varies between medical conditions. As this systematic review shows promising results for polypills, research could be extended to a wider range of medical conditions, populations, and health systems, as well as beyond high-income countries.

## Figures and Tables

**Figure 1 pharmaceutics-12-00190-f001:**
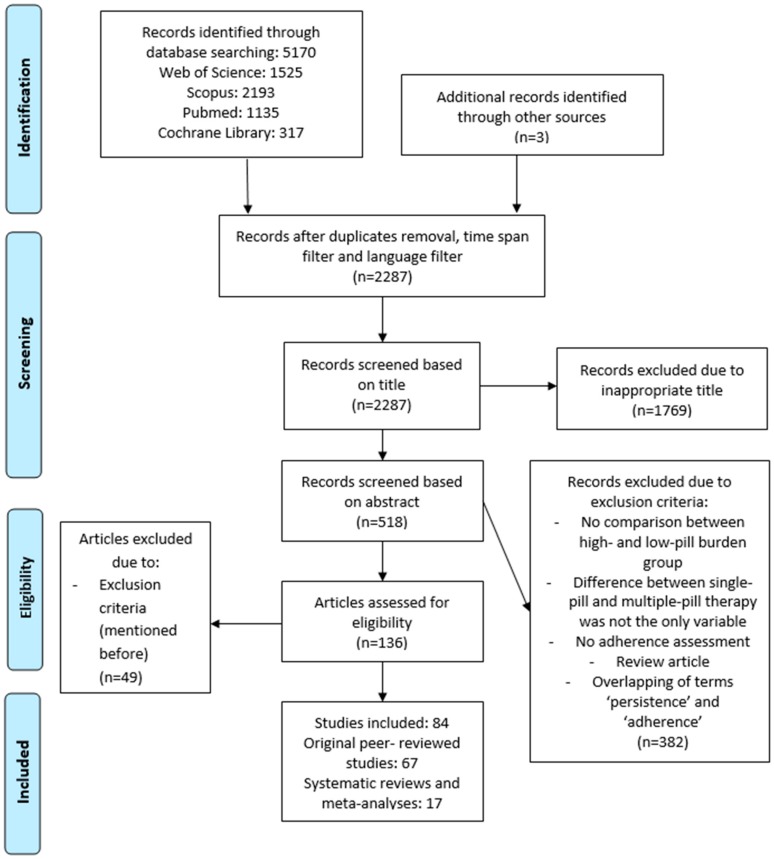
Preferred Reporting Items for Systematic Reviews and Meta-Analyses (PRISMA) flow chart of the article selection.

**Figure 2 pharmaceutics-12-00190-f002:**
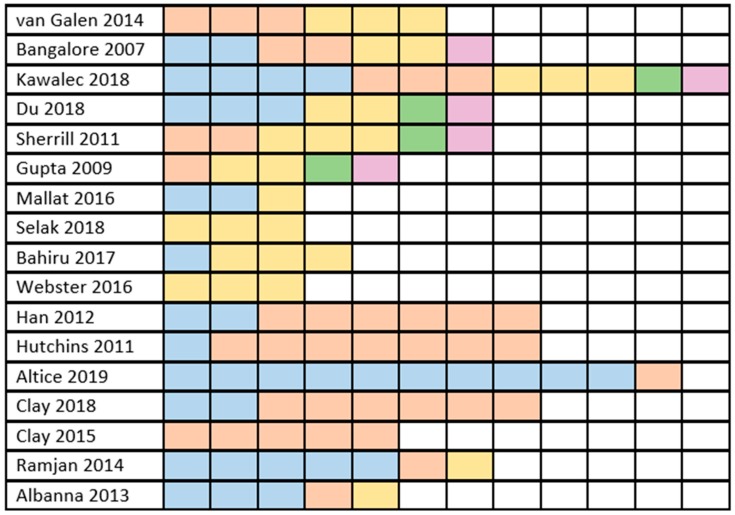
Visual representation of the overlap of the studies included in other systematic reviews and meta-analyses (SR and MA; *n* = 17). One row represents one SR/MA. Each colored square symbolizes one article, and the number of colored squares is equal to the number of studies included in the corresponding SR/MA. Different colors represent into how many SRs/MAs an article was included (e.g., if all the colored squares in a row are blue, all the articles are unique to only this SR/MA). Blue: study included only in one SR/MA. Red: study included in two different SRs/MAs. Yellow: study included in three different SRs/MAs. Green: study included in four different SRs/MAs. Purple: study included in five different SRs/MAs.

**Figure 3 pharmaceutics-12-00190-f003:**
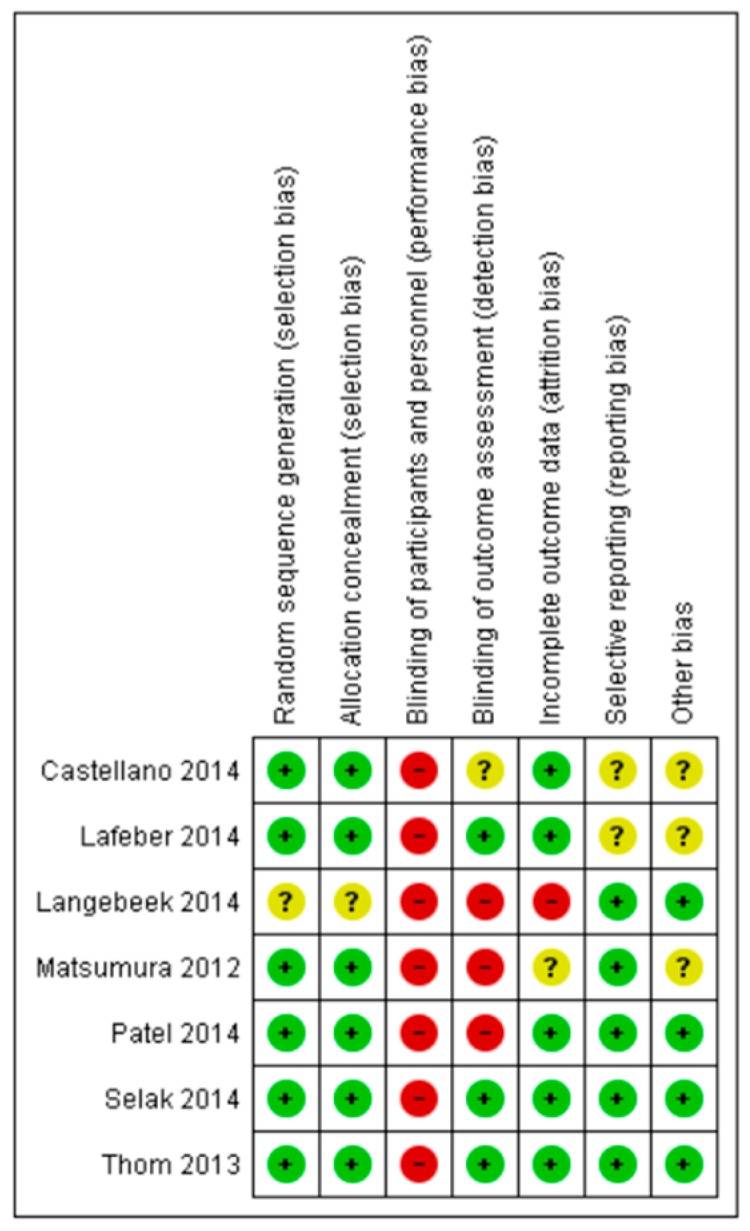
Results of risk of bias assessment for all randomized controlled trials (RCTs). Green: low risk of bias; red: high risk of bias; yellow: unclear risk of bias.

**Figure 4 pharmaceutics-12-00190-f004:**
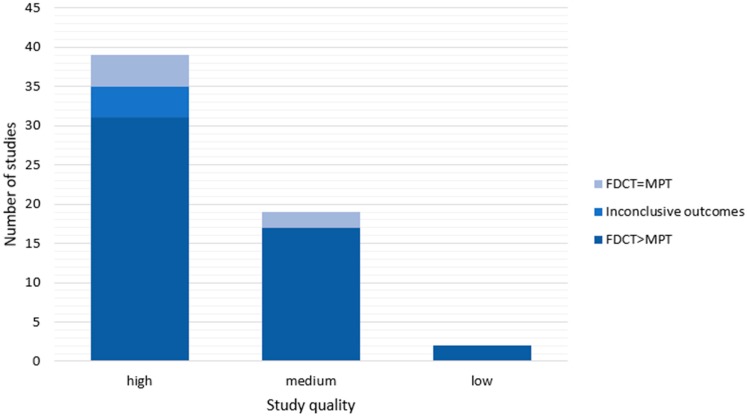
Number of studies with a certain outcome per study quality. FDCT, fixed-dose combination therapy; MPT, multipill therapy. Inconclusive outcomes: see Table 5, Table A2 (Appendix B), and Table A3 (Appendix C) for additional information.

**Table 1 pharmaceutics-12-00190-t001:** Summary of conclusions per disease in previously published systematic reviews and meta-analyses (*n* = 17).

Disease	Conclusions Concerning Adherence to FDCT	Study Design [Reference]		
		Meta-Analysis	Systematic Review with Meta-Analysis	Systematic Review
Various diseases (*n* = 2)	FDCT > MPT (*n* = 1)	[13]		
	Inconclusive (*n* = 1)		[29]	
Hypertension (*n* = 5)	FDCT > MPT (*n* = 4)	[14,15,16]	[25]	
	Inconclusive (*n* = 1)		[26]	
CVD (*n* = 3)	FDCT > MPT (*n* = 2)	[18]		[21]
	FDCT = MPT (*n* = 1)	[17]		
HIV (*n* = 4)	FDCT > MPT (*n* = 4)	[19,20]	[24,27]	
Diabetes (*n* = 2)	FDCT > MPT (*n* = 2)		[28]	[22]
Tuberculosis (*n* = 1)	FDCT not favored (*n* = 1)		[23]	

FDCT, fixed-dose combination therapy; MPT, multipill therapy; CVD, cardiovascular disease; HIV, human immunodeficiency virus.

**Table 2 pharmaceutics-12-00190-t002:** General information about reviewed articles (*n* = 67).

Information of Interest	Result (Number of Studies with a Certain Feature)	References
**Disease**	CVD (*n* = 11)	[35,45,48,49,52,58,59,60,61,62,63]
	HT (*n* = 31)	[5,33,38,39,40,41,42,46,53,55,56,64,65,66,67,68,69,70,71,72,73,74,75,76,77,78,79,80,81,82,83]
	DMII (*n* = 10)	[31,32,36,43,44,51,54,84,85,86]
	HIV (*n* = 14)	[30,34,37,47,50,57,87,88,89,90,91,92,93,94]
	LUTS/BHP (*n* = 1)	[95]
**Country where the study was conducted (in alphabetical order)**	Argentina (*n* = 1)	[35]
	Australia (*n* = 3)	[45,58,61]
	Austria (*n* = 1)	[66]
	Belgium (*n* = 2)	[66,88]
	France (*n* = 1)	[59]
	Germany (*n* = 4)	[7,66,67,79]
	Greece (*n* = 1)	[85]
	India (*n* = 1)	[52]
	Ireland (*n* = 1)	[52]
	Italy (*n* = 8)	[35,40,42,69,70,84,87,92]
	Japan (*n* = 2)	[73,75]
	Korea (*n* = 1)	[64]
	The Netherlands (*n* = 5)	[52,59,66,88,95]
	New Zealand (*n* = 1)	[48]
	Paraguay (*n* = 1)	[35]
	Romania (*n* = 1)	[77]
	Spain (*n* = 2)	[35,90]
	Switzerland (*n* = 1)	[66]
	Taiwan (*n* = 5)	[5,53,71,78,80]
	UK (*n* = 1)	[52]
	USA (*n* = 36)	[30,31,32,33,34,36,37,38,39,41,43,44,46,49,50,51,54,55,56,57,60,62,63,65,68,72,74,76,81,82,83,86,89,91,93,94]
**Follow-up period (given as the exact, average, or minimal value, depending on the study)**	6 weeks (*n* = 1)	[91]
	2 months (*n* = 4)	[37,57,89,93]
	3 months (*n* = 1)	[73]
	18 weeks (*n* = 1)	[59]
	24 weeks (*n* = 1)	[85]
	6 months (*n* = 16)	[30,32,42,43,44,46,51,54,55,58,66,68,70,72,75,87]
	9 months (*n* = 1)	[35]
	12 months (*n* = 26)	[31,33,36,38,39,40,41,48,49,52,56,60,62,64,65,67,71,74,76,80,81,82,83,84,86,95]
	15 months (*n* = 1)	[78]
	18 months (*n* = 3)	[34,45,50]
	1.7 years (*n* = 1)	[94]
	96 weeks (*n* = 1)	[90]
	24 months (*n* = 5)	[5,53,61,69,88]
	33 months (*n* = 1)	[47]
	36 months (*n* = 1)	[63]
	4 years (*n* = 1)	[77]
	5 years (*n* = 2)	[79,92]
**Year of publication**	2002–2004 (*n* = 3)	[32,43,49]
	2005–2007 (*n* = 4)	[40,54,60,89]
	2008–2010 (*n* = 15)	[30,33,36,38,39,41,44,46,51,55,56,62,68,72,87]
	2011–2013 (*n* = 10)	[31,34,37,52,63,65,73,75,83,84]
	2014–2016 (*n* = 22)	[5,35,42,45,47,48,50,57,58,59,66,70,76,78,80,81,82,85,86,88,91,93]
	2017–2019 (*n* = 13)	[14,53,61,64,69,71,74,77,79,90,92,94,95]
**Study design**	Randomized clinical study (*n* = 7)	[35,45,48,52,59,75,88]
	Retrospective cohort study (*n* = 52)	[5,31,32,33,36,37,38,39,40,41,42,43,44,46,47,49,50,51,53,54,55,56,57,58,60,61,62,63,64,65,67,68,69,70,71,72,74,76,77,78,79,80,81,82,83,84,86,89,92,93,94,95]
	Prospective cohort study (*n* = 8)	[30,34,66,73,85,87,90,91]
**Number of drugs in the polypill examined in the study**	Two drugs (*n* = 41)	[5,31,32,33,36,38,39,40,41,42,43,44,46,49,51,53,54,55,56,58,60,61,62,63,64,65,67,68,69,70,71,72,73,75,78,79,80,83,85,89,95]
	Three drugs (*n* = 11)	[30,34,35,37,47,50,66,81,82,87,88]
	Four drugs (*n* = 5)	[45,48,52,59,90]
	Five drugs (*n* = 1)	[57]
	Not mentioned (*n* = 9)	[74,76,77,84,86,91,92,93,94]

CVD, cardiovascular disease; DMII, diabetes mellitus type 2; LUTS/BHP, lower urinary tract symptoms associated with benign prostatic hyperplasia; HT, hypertension; HIV, human immunodeficiency virus; UK, United Kingdom; USA, United States of America.

**Table 3 pharmaceutics-12-00190-t003:** Visualization of number of active ingredients contained in a polypill. Written in the table are numbers of the studies with the given characteristics (disease and number of active ingredients in the polypill).

Disease	Number of Studies Dealing with a Polypill Containing a Given Number of Active Ingredients (2, 3, 4, 5, or Not Mentioned)				
	II	III	IV	V	Not mentioned
CVD	6 [49,58,60,61,62,63]	1 [35]	4 [45,48,52,59]	0	0
HT	25 [5,33,38,39,40,41,42,46,53,56,64,65,67,68,69,70,71,72,73,75,78,79,80,83,94]	3 [66,81,82]	0	0	3 [74,76,77]
DMII	8 [31,32,36,43,44,51,54,85]	0	0	0	2 [84,86]
HIV	1 [89]	7 [30,34,37,47,50,87,88]	1 [90]	1 [57]	4 [91,92,93,94]
LUTS/BPH	1 [95]	0	0		0
Sum	41	11	5	1	9

CVD, cardiovascular disease; DMII, diabetes mellitus type 2; LUTS/BHP, lower urinary tract symptoms associated with benign prostatic hyperplasia; HT, hypertension; HIV, human immunodeficiency virus.

**Table 4 pharmaceutics-12-00190-t004:** Methods for measuring adherence applied in the articles (*n* = 67).

Method	Study-Specific/General	Short Description	Assessment of Level of Adherence	*n* of Studies [References]
Medication possession ratio (MPR)	General	Uses pharmacy prescription claims calculated as the number of days’ supply divided by the number of days between the first refill and the end of the follow-up period	Low adherence: MPR < 0.5 Intermediate adherence: MPR = 0.5–0.8 High adherence: MPR > 0.8	*n* = 30 [5,31,32,33,36,39,40,41,43,44,49,50,51,54,55,57,58,60,62,63,64,67,80,82,84,86,89,91,94,95]
Proportion of days covered (PDC)	General	Uses prescription claims data; every day has to be covered by the medication; coverage is calculated based on the refill data. For example, if the patient has 30 pills in his prescription (1/day) and he gets a refill after 40 days, his PDC is 30/40 or 75%.	A PDC of >80% is considered adherent.	*n* = 21 [38,42,46,53,56,61,65,68,69,70,71,72,74,76,78,79,81,82,83,90,93]
Pill count	General	Healthcare professional pays an unexpected visit to the patient’s home and counts the pills left; difference between the number of pills dispensed and the number of pills not taken gets divided by number of prescribed pills.	Patient is considered adherent, if the percentage is between 80% and 110%.	*n* = 6 [30,35,47,73,75,91]
Morisky scale	General	Questionnaire containing eight questions; a self-assessment scale.	Based on the sum of the scores.	*n* = 3 [35,59,66]
Self-reporting	Study-specific	1. Asking the patients about the names and dosages of all drugs that are currently taken [48]. 2. Self-reported use of indicated combination treatment (antiplatelet, statin, and ≥2 blood-pressure-lowering therapies for ≥4 of the previous seven days) [45,52]. 3. Self-reporting of missed doses at each medical visit [50]. 4. Completing a compliance questionnaire—nine questions about the names and dosages of all drugs, missing doses, treatment interruptions, etc. [85]	1. Adherent: patients reported taking an antiplatelet, statin, and two or more blood-pressure-lowering drugs. Non-adherent: patients who forgot one or more drugs. 2. Level of adherence not assessed. 3. It was assumed that each day of ART missed was an additional day between refills of a 30-day supply → MPR method was applied. 4. Adherent: not missing any drug dose or no more than 2 doses per week, received the correct dosage of the medication, and not interrupting their treatment.	*n* = 5 [45,48,50,52,85]
Visual Analog Scale (VAS)	General	Uses information given by the patient who performs self-assessment of adherence on a scale 0–100.	Non-adherent: 0 Perfectly adherent: 100	*n* = 2 [34,87]
Simplified Medication Adherence Questionnaire (SMAQ)	General	Self-reported questionnaire focused on HIV patients, containing six items.	Method of assessment is not given in the article.	*n* = 1 [88]
Prescription records review	Study-specific	Computing the total number of consecutive months that was covered by antihypertensive prescriptions during the study; adherence is expressed as percentage of time.	Low adherence: <20% Medium adherence: 20–79% High adherence: ≥80%	*n* = 1 [77]
Electric adherence monitoring	General principle, study-specific design (depends on the dosage form, dosage regimen, etc.)	The medication vial was closed with a cap containing a microprocessor, which was recording date and time of all openings. The vial was filled with the exact amount of medication required for the complete treatment period. The participant was instructed not to open the vial except when taking the medication according to the prescribed regimen.	Based on whether the patient was taking the doses daily and according to the schedule.	*n* = 1 [59]
Time to the first instance to discontinuation *	General method, study-specific definition	Defined as no repeat of prescription within 150% of the previous days’ supply.	Treatment discontinuation: break of therapy for more than 150% of the previous days’ supply.	*n* = 1 [79]
RDD/PDD ratio	General	Ratio between received daily dose (corresponds to the ratio between total doses received and treatment days) and prescribed daily dose (stands for the intention to treat and the real prescriptive tendency).	Adherence is assessed and given only as an RDD/PDD ratio; there is no evaluation of what is considered high or low adherence.	*n* = 1 [92]

* Usually used as a measure of therapy persistence. ART, antiretroviral therapy; RDD, received daily dose; PDD, prescribed daily dose.

**Table 5 pharmaceutics-12-00190-t005:** Summary of the study results per disease. Statistically significant differences in adherence outcomes are presented and considered.

Disease	Comparison of Adherence Outcome between FDCT and MPT; Number of Studies with Certain Result Is Given in Parenthesis	References
CVD (*n* = 11)	FDCT > MPT (*n* = 9)	[35,45,48,49,52,59,60,62,63]
	FDCT = MPT (*n* = 1)	[58]
	Inconclusive * (*n* = 1)	[61]
HT (*n* = 31)	FDCT > MPT (*n* = 28)	[5,33,38,39,40,41,42,46,53,55,56,64,65,66,67,68,69,70,71,72,73,74,77,78,79,81,82,83]
	FDCT = MPT (*n* = 2)	[75,76]
	Inconclusive * (*n* = 1)	[80]
DMII (*n* = 10)	FDCT > MPT (*n* = 9)	[31,32,36,44,51,54,84,85,86]
	Inconclusive * (*n* = 1)	[43]
HIV (*n* = 14)	FDCT > MPT (*n* = 10)	[30,37,57,87,88,89,91,92,93,94]
	FDCT = MPT (*n* = 3)	[34,50,90]
	Inconclusive * (*n* = 1)	[47]
Other (*n* = 1)	FDCT = MPT (*n* = 1)	[95]

* Several outcomes were observed (FDCT < MPT or FDCT > MPT or FDCT = MPT). See Table A2 (Appendix B) and Table A3 (Appendix C) for additional information. FDCT, fixed-dose combination therapy; MPT, multipill therapy; CVD, cardiovascular disease; DMII, diabetes mellitus type 2; LUTS/BHP, Lower urinary tract symptoms associated with benign prostatic hyperplasia; HIV, human immunodeficiency virus.

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
