# Peer review of "Does the Polypill Improve Patient Adherence Compared to Its Individual Formulations? A Systematic Review"

_pharmaceutics, 2020, doi:10.3390/pharmaceutics12020190_

Round 1

Reviewer 1 Report

In this paper this systematic review on influence of polypill on patient adherence is given. The paper is well organized and prepared. The only objection is that paper is too long. The aim of Journal is to encourage scientists to publish their experimental results and theoretical assumptions in as much detail as possible (there is no restriction on the length of the papers). Therefore, I do not consider that the authors should shorten their paper.   

Author Response

Thank you very much for your comments. As the aim of our study was to make a comprehensive systematic review that would cover a wide spectrum of articles, we realise that the length of the paper could be an issue. However, the core sections of the article only cover 17 out of total 67 pages and the rest is characterised as appendices and references, which would probably be more applicable to the more interested readers. Thank you for accepting our paper regardless of its length.

Reviewer 2 Report

The systematic review submitted by Ana Baumgartner et al. is a valuable and comprehensive work dealing with the problem of patients' adherence to the recommended pharmacotherapy.

In my opinion, the study has been well planned and performed. The results are clearly presented and discussed. My only concern regards Figure 2. which is unclear and should be better explained.

The authors addressed the advantages and limitations of the study properly. In my opinion, the manuscript can be accepted for publication after minor corrections.

Author Response

Thank you very much for your comments. Please see our responses below.

Point 1: My only concern regards Figure 2, which is unclear and should be better explained.

Response 1: Thank you for this comment, we appreciate your concern. We revised the caption of Figure 2 accordingly (please see the attachment, page 5, lines 152–160) and hope that the figure is more comprehensible in the present form. If there is still a problem with understanding the figure, we would be happy to give further clarifications.

Reviewer 3 Report

The reviewer has comments and suggestions as follows:

  1. It would be better to write the strength of your study at the first paragraph in discussion session rather than simply describing the list of results. Such change may attract the interests of the readers more.
  2. It appears that your analyses are heavily focused on the effect of polypill on the adherence. If you like to insist that polypill could improve the clinical outcome, the reviewer recommend additional analyses with appropriate references to support your claim.
  3. The reviewer recommends also stating the disadvantage of polypill in clinical settings in addition to the advantage to be unbiased and misled the readers.
  4. Your analysis covered mainly on four medical conditions (CVD, HT, DM2, and HIV), which does not appear to be sufficient to generalize that polypill is better for the other conditions. Please revise the manuscript accordingly.

Author Response

Thank you very much for reading the paper and your comments. Please see our point-by-point responses below.

Point 1: It would be better to write the strength of your study at the first paragraph in discussion session rather than simply describing the list of results. Such change may attract the interests of the readers more.

Response 1: Thank you for the comment. We took it into consideration and added a short section on the strength of our study in the beginning of the discussion (please see page 14, lines 254–256).

Point 2: It appears that your analyses are heavily focused on the effect of polypill on the adherence. If you like to insist that polypill could improve the clinical outcome, the reviewer recommend additional analyses with appropriate references to support your claim.

Response 2: We appreciate your concern. However, the aim of our study was to limit our focus on patient adherence and its potential connection with pill burden. Therefore, we realise we cannot claim that there is definitely a connection between polypills and clinical outcome, which is also mentioned in the last paragraph pf the discussion (please see page 16, lines 369–381). We believe that for evaluating the role of polypills in achieving better clinical outcomes, another systematic review could be conducted.

Point 3: The reviewer recommends also stating the disadvantage of polypill in clinical settings in addition to the advantage to be unbiased and misled the readers.

Response 3: Thank you very much for your concern. We took it into consideration and decided to write a short paragraph that would discuss the drawbacks of the polypills and thereby prevent misleading the readers (please see page 16, lines 361–368).

Point 4: Your analysis covered mainly on four medical conditions (CVD, HT, DM2, and HIV), which does not appear to be sufficient to generalize that polypill is better for the other conditions. Please revise the manuscript accordingly.

Response 4: Thank you very much for your comment, we acknowledge the importance of this issue. A clarification for the interpretation of the results has been added (please see lines 257–258 and 370–371). It should also be acknowledged that were no restrictions concerning medical conditions in our research. However, these are the medical conditions where most research on connection between polypills and adherence has been done. Also, we have already addressed the issue in several sections of the discussion (please see lines 286–301, 317–320). Should you have any further concerns regarding this issue, we would be happy to further revise our paper.
